# Innate Lymphoid Cell Plasticity in Mucosal Infections

**DOI:** 10.3390/microorganisms11020461

**Published:** 2023-02-12

**Authors:** Anna A. Korchagina, Ekaterina Koroleva, Alexei V. Tumanov

**Affiliations:** Department of Microbiology, Immunology and Molecular Genetics, University of Texas Health Science Center at San Antonio, 7703 Floyd Curl Dr., San Antonio, TX 78229, USA

**Keywords:** innate lymphoid cells, ILC plasticity, mucosal pathogens

## Abstract

Mucosal tissue homeostasis is a dynamic process that involves multiple mechanisms including regulation of innate lymphoid cells (ILCs). ILCs are mostly tissue-resident cells which are critical for tissue homeostasis and immune response against pathogens. ILCs can sense environmental changes and rapidly respond by producing effector cytokines to limit pathogen spread and initiate tissue recovery. However, dysregulation of ILCs can also lead to immunopathology. Accumulating evidence suggests that ILCs are dynamic population that can change their phenotype and functions under rapidly changing tissue microenvironment. However, the significance of ILC plasticity in response to pathogens remains poorly understood. Therefore, in this review, we discuss recent advances in understanding the mechanisms regulating ILC plasticity in response to intestinal, respiratory and genital tract pathogens. Key transcription factors and lineage-guiding cytokines regulate this plasticity. Additionally, we discuss the emerging data on the role of tissue microenvironment, gut microbiota, and hypoxia in ILC plasticity in response to mucosal pathogens. The identification of new pathways and molecular mechanisms that control functions and plasticity of ILCs could uncover more specific and effective therapeutic targets for infectious and autoimmune diseases where ILCs become dysregulated.

## 1. Introduction

Mucosal surfaces represent protective barriers which are continuously exposed to external factors such as small particles, commensal bacteria, antigens from diet, inhaled air, and pathogens. In an environment where mucosal surfaces encounter a lot of foreign antigens, the host immune system uses highly regulated mechanisms to maintain tissue homeostasis and to recognize pathogenic microorganisms. Pathogen invasion of mucosal tissues can be recognized by pattern recognition receptors on epithelial and innate immune cells leading to their rapid activation to induce immune response [1,2]. Upon pathogen invasion, innate immune cells such as macrophages and dendritic cells produce chemokines and cytokines, which determine activation of particular immune cells and type of immune response within the tissue [2]. Innate lymphoid cells (ILCs) are an essential part of innate mucosal immunity. ILCs can sense environmental changes and rapidly respond by producing effector cytokines to limit pathogens spread, initiate tissue recovery, regulate epithelial cell differentiation and activate other immune cells, thereby facilitating inflammatory response [3,4,5]. Moreover, recent studies reported that ILCs can regulate the activity of adaptive immune cells through direct cell-to-cell contact and cytokine production [3,6,7,8,9,10]. ILCs mimic their T cell counterparts in terms of cytokine production and function [10,11,12]. Although T cells are critical for specific immune response against pathogens, their differentiation and polarization require time and can take days and weeks after infection. In contrast to T cells, ILCs can be rapidly activated after pathogen invasion. ILCs are classified into five distinct subsets based on their developmental pathways, lineage-determining transcription factors (LDTF) and effector functions: ILC1, ILC2, ILC3, lymphoid tissue inducer (LTi) cells and natural killer (NK) cells [13]. The ILC differentiation program is under the control of LDTFs which regulate the expression of subset-defining genes such as signature cytokines, chemokines and receptors. LDTFs are critical not only for ILC development but also for guiding their phenotype and effector functions [11,14,15]. Thus, the transcription factor Eomes controls NK cells differentiation, while T-bet, GATA3 and RORγt are important for ILC1, ILC2 and ILC3 development and their effector cytokine production, respectively [13,16]. Intracellular pathogens and viruses activate ILC1s to produce IFNγ and TNF [5,17]. Along with these effector cytokines, NK cells can mediate cytotoxic activity by releasing perforins and granzymes to induce apoptosis of target cells [18,19]. In contrast, ILC2s respond to parasites or allergens by producing IL-4, IL-5, IL-9 and IL-13 [16]. Additionally, ILC2s release amphiregulin to promote repair of intestinal and airway epithelial cells [20,21]. ILC3s respond to extracellular pathogens and fungi by producing IL-22 and IL-17 to maintain epithelial cell functions and promote tissue repair [13,22], and can produce GM-CSF to activate myeloid cells in the gut [8,23,24,25]. Extensive research over the last decade revealed plasticity within all ILC subsets that is largely controlled by tissue-derived cytokines and specific cytokine receptors on ILCs [26,27,28]. Activation of cytokine receptors on ILCs leading to induction of key transcription factors drives transdifferentiation of ILC subsets. Distinct one-way and two-way plasticity have been described in both mouse and human ILCs: NK→ILC1, ILC3↔ILC1, ILC2↔ILC1, ILC3↔ILC2 [12,26,29,30,31] (Figure 1).

NK→ILC1 plasticity is driven by IL-12 and TGF-β, resulting in upregulation of T-bet and downregulation of Eomes [32,33,34]. ILC3→ILC1 plasticity is driven by IL-12 and IL-1β, leading to downregulation of RORγt and upregulation of T-bet [12,29,35,36]. T-bet is critical for IFNγ production, which is necessary for protection against intracellular pathogens [5,17]. Conversely, ILC1→ILC3 plasticity is driven by IL-23, IL-1β, and retinoic acid (RA), which leads to upregulation of RORγt and downregulation of T-bet [29]. Transcription factors c-Maf, Batf and Zbtb46 further stabilize ILC3 lineage [37,38,39,40]. ILC3s in turn produce IL-22 and IL-17 which are critical for protection against extracellular bacteria and fungi [13,41]. Similar to the ILC3→ILC1 transition, ILC2→ILC1 plasticity is driven by IL-12 and IL-1β which leads to downregulation of GATA3 and upregulation of T-bet [30,31,42,43,44]. Conversely, the reversed ILC1→ILC2 plasticity is induced by IL-4 which leads to GATA3-dependent expression of type 2 cytokines IL-4, IL-13, IL-5 to promote anti-parasitic immunity [42,45,46]. Additionally, ILC2→ILC3 plasticity is driven by production of IL-1β, IL-23 and TGF-β, which leads to upregulation of RORγt expression and consequent downregulation of GATA3 [46]. IL-4 also promotes the reversed ILC3→ILC2 plasticity by inducing GATA3 expression [45]. Thus, mucosal pathogens or changes in environmental conditions lead to activation of numerous intracellular signaling pathways that induce production of cytokines, which in turn promote changes in ILC phenotypes and their functions [26,29]. Therefore, ILC plasticity allows tissue resident cells to quickly adjust to changes upon pathogen invasion to promote distinct types of immune responses at different stages of disease. At the same time, it is possible that pathogens can exploit the plasticity of ILCs to avoid host protective responses. Accumulating evidence suggests that ILC transdifferentiation can be reversed, underlying the existence of mechanisms balancing ILC composition under physiological conditions and pathogen invasion to prevent excessive inflammation. It is now evident that ILC plasticity is not only an important driver of protective immune responses but can also lead to exacerbation of various chronic and inflammatory diseases [5,12,42,46,47]. However, the molecular mechanisms of ILC plasticity and its impact on immune response to mucosal pathogens remain poorly understood. Therefore, in the next sections of this review we will discuss the emerging role of ILC plasticity in response to pathogens (Table 1).

## 2. Role of Tissue Microenvironment in ILC Plasticity

The tissue-specific microenvironment instructs ILCs to acquire different fates to sustain homeostatic conditions or efficiently resolve inflammation that could be realized through several mechanisms such as differentiation from ILC precursors, migration or plasticity [70,71]. Both environmental factors (cytokines, dietary metabolites, microbiota) as well as intrinsic factors, such as transcription factors, can influence ILC composition within the tissue [14,15,35,51,70,72,73,74]. Single cell transcriptomic analysis of small intestine revealed 13 transcriptional states of three main ILC subsets (ILC1, ILC2 and ILC3) and two additional transcriptional clusters that are district from ILC subsets [75]. Under homeostatic conditions, conventional ILC1s were clustered into four distinct transcriptional profiles and displayed gradient expression of T-bet [75]. Interestingly, within these four ILC1 groups one displayed expression of ILC2 signature cytokines and transcription factors, suggesting that under physiological conditions ILC2s may transdifferentiate to ILC1s [75]. Compartmentalization of ILC2 subset showed four different transcriptional states that exhibited distinct expression of GATA3 as well as IL-5 and amphiregulin [75]. ILC3 subset was subdivided into five transcriptional states highlighting the heterogeneity of ILC3s. Interestingly, among two additional transcriptional clusters that were district from main ILC clusters, one had expression of both IFNγ and RORγt as well as NKp46 (natural cytotoxicity receptor-NCR) [75], suggesting that this cluster may represent ILC3 subset, transdifferentiated to ILC1s. In line with this study, ILC subsets with mixed ILC3-ILC1 signature gene expression profiles were found in human tonsils [76]. Identification of these intermediate transcriptional phenotypes of ILCs demonstrate that ILCs can acquire distinct effector functions and phenotypes depending on tissue localization and specific environmental cues. Consistent with this, it was found that mouse ILC3s regardless of their origin can migrate to different tissues where they change phenotype based on the tissue microenvironment [77]. Furthermore, another study characterized ILC heterogeneity in distinct anatomical sites such as lymphoid and mucosal tissues in humans under homeostatic conditions [78]. It was found that expression of NCRs by ILCs is more heterogeneous in mucosal tissues and spleen compared to the lungs and adipose tissue, supporting the data obtained in mouse models [77,78,79]. Analysis of transcriptional profiles of ILCs from spleen, lungs and intestine revealed that the ILC3 subset is the most transcriptionally distinct, whereas the transcriptional profile of ILC1s overlapped with intestinal and splenic NK cells under physiological conditions [78,80]. Thus, these studies suggest that ILCs have a high degree of functional tissue-dependent specialization within the canonical ILC subsets and that tissue microenvironment impacts ILC phenotypes. However, little is known of how the tissue-specific heterogeneity of ILCs impacts the response to pathogens.

## 3. ILC Plasticity in Response to Gut Infections

The gastrointestinal tract (GI) is constantly exposed to dietary antigens and commensals. Numerous innate and adaptive immune cells, highly represented in the intestine constantly adapt to the changing environment by modulating their phenotype and functions. Although similar immune cell populations can be found in upper and lower GI tract, the balance and phenotypes of the immune cells vary and depend on the physiological function of the intestinal region [81]. Furthermore, the distribution of commensal bacteria in the gut can affect the immune cell composition in a particular gut region [82]. Hence, under homeostatic conditions Tregs, which regulate immune tolerance against commensals, are enriched in the colon whereas the small intestine is populated by numerous IL-17 producing CD4^+^ T cells [81,83]. Accordingly, microbiota loads are increased from the proximal to the distal parts of the small intestine, with the highest abundancy of microbiota in the colon [82]. ILCs are present throughout the gut with increased abundancy of ILC3s in the lower intestine in both humans and mice [35,66,84]. During inflammation or pathogen invasion ILCs can undergo plasticity in the intestine. ILC plasticity was first described in the mouse small intestine where plastic changes of NKp46^+^ ILC3s led to production of IFNγ [35]. Lately, it was shown that NCR^+^ ILC3s may participate in host defense against *Salmonella* infection [51] as ILC3s in the small intestine can convert their phenotype into IFNγ-producing ILC1s (Figure 2). Although NKp46^+^ ILCs are the main innate source of IFNγ production in the small intestine during *Salmonella* infection [51], accumulating data indicate that NCR^−^ ILCs can also produce IFNγ under inflammatory settings. For example, in response to *Yersinia enterocolitica* infection, NCR^−^ ILCs produce IFNγ to initiate protective immune response in the small intestine [61]. In line with this study, another study demonstrated that NK1.1^−^ ILC1s are the main innate source of IFNγ in the colon during *Campylobacter jejuni* (*C. jejuni*) infection [52]. Moreover, *C. jejuni* infection induces conversion of NK1.1^−^ ILC3s to IFNγ-producing NK1.1^−^ ILC1s in the colon [52] (Figure 2). 

The differential capacity of NCR^+^ ILCs and NCR^−^ ILCs to produce IFNγ could be a result of exposure to specific environmental factors that are present in the small and large intestine. Constant exposure to these factors may facilitate phenotypic and functional changes in ILCs. Thus, the environment in the small intestine may promote expression of NCR by ILCs while in the colon the ratio of NCR^+^ cells to NCR^−^ cells is lower [25]. Cell-fate mapping of the intestinal ILCs identified NCR^−^ ILC3s with a prior history of NCR expression [85,86]. Transcriptomic analysis revealed the population of NCR^−^ ILC3s with low expression of CCR6 and NCR1, which may represent an intermediate population between NCR^−^ and NCR^+^ ILC3 [86]. 

TGF-β and Notch are key regulators of NCR^+^ to NCR^−^ ILC3 plasticity [86]. Notch promotes differentiation of NCR^−^ ILC3s to NCR^+^ ILC3s, whereas TGF-β reverses this transition [86]. Loss of TGF-β signaling resulted in increased numbers of NCR^+^ ILC3s in the small intestine indicating that TGF-β can promote the conversion of NCR^+^ ILC3 to NCR^−^ ILC3s [86]. TGF-β also regulates proliferation and differentiation of immune cells such as T cells, NK cells and DCs [87]. More recently, it was shown that TGF-β in combination with IL-6 can enhance expression of Batf in ILC3s to maintain ILC3 phenotype by suppressing the expression of T-bet [37]. The development of NCR^+^ ILC3s from NCR^−^ ILC3s is dependent on T-bet and partially on Notch signaling as NCR^+^ ILC3s are drastically reduced in mice with abrogated T-bet or Notch signaling [51,85,88]. Notch and downstream aryl hydrocarbon receptor (Ahr) signaling plays a critical role in ILC3 development [85,88]. Additionally, Notch signaling promotes ILC2 to ILC3 plasticity by inducing differentiation of natural ILC2s to IL-17 and IL-13 producing inflammatory ILC2 populations via upregulation of RORγt expression [89].

Ahr expression is regulated by the gut-specific environment and helps to maintain homeostatic ILC2-ILC3 balance [90]. Among all ILCs in the gut, ILC2s have the highest Ahr expression, which leads to suppression of ST2 and effector molecules IL-5, IL-13 and amphiregulin [90]. Moreover, pharmacological activation of Ahr results in enhanced ILC3-mediated immune response with reduced functional activity of ILC2s, suggesting that Ahr can activate different types of immune responses depending on type of the pathogen [90]. In addition to controlling ILC2-ILC3 balance, Ahr promotes IL-7 receptor (IL-7R) expression in ILC3s which is important for cell survival and for stabilization of RORγt expression in intestinal NCR^+^ ILC3s [35,91]. Given that Ahr and Notch deficiencies lead to reduction of NCR^+^ ILC3 [51,85,88], it is possible that Ahr can promote ILC3 to ILC1 plasticity in response to mucosal pathogens. However, this possibility has not been tested yet.

ILC3s are severely decreased in circulation and intestinal mucosa of HIV patients and this depletion correlated with disease progression [66,67,68,92]. This ILC3 reduction is potentially caused by high level of type I IFN production after HIV infection by plasmacytoid dendritic cells (pDCs), which induces Fas (CD95) expression on ILC3s thereby promoting apoptosis [67]. It is widely accepted that chronic inflammation in HIV patients can be caused by intestinal barrier dysfunction and translocation of intestinal microbes from gastrointestinal tract into circulation [66,69]. A reduced number of IL-22 producing ILC3s and Th22 cells was found in the gut mucosa of chronic HIV-infected patients and in SIV-infected Chinese macaques [66,69], suggesting that disruption of the gut barrier is mediated by impaired function of IL-22 producing cells. Furthermore, loss of ILCs in the gut was associated with increased neutrophil numbers in lamina propria, type I IFN production and expansion of inflammatory TCF7^+^ NK cells [93]. However, plasticity among NK and ILC subsets in the pathogenesis of HIV-induced intestinal disease needs to be further investigated. 

Overall, ILC plasticity in the gut during pathogen invasion can be regulated at multiple levels, including microbiota-driven changes, hypoxia and changes driven by pathogens themselves. Thus, under homeostatic conditions, the microbiota stimulates production of IL-7 by epithelial cells to maintain ILC3s [35,94]. These cells, in turn, promote production of antimicrobial peptides by epithelial cells to restrain microbiota in the lumen [13]. Additionally, ILC3s maintain T cell tolerance to microbiota through MHCII expression to prevent T cell-driven intestinal inflammation [6,40,95]. On the other hand, IL-7 promotes c-Maf induction in ILC3s, which in turn enhances IL-7R expression to support RORγt expression by NCR^+^ ILC3s under homeostatic conditions [35,39]. However, pathogen invasion shifts the balance towards proinflammatory cytokines such as IL-12, IL-18, IL-1β, IL-23, resulting in ILC3 plasticity. In addition to IL-7R regulation, c-Maf inhibits T-bet expression in NCR^+^ ILC3s therefore changing the ILC3/ILC1 balance in the intestine [39]. Moreover, c-Maf can restrain transition to an ILC1-like phenotype independently from microbiota and adaptive immune cells [39,96]. Finally, HIF-1α, another potential regulator of ILC plasticity, can change microbiome composition [97], potentially leading to ILC plasticity. The role of HIF-1α in hypoxia-driven ILC plasticity is further described in Section 5 below. Further studies are needed to fully understand the complex interactions between pathogens, microbiota and host cells leading to ILC plasticity. 

## 4. Role of Gut Microbiota in ILC Plasticity

The outcome of enteric infections depends on the host microbiota which can either enhance protection by preventing pathogen invasion or exacerbate inflammation and tissue damage. Enteric pathogens compete with metabolically similar microbiota for nutrients [98]. For example, mucosal carbohydrate availability is dependent on microbiota and can influence bacterial growth and prevent luminal colonization by *Salmonella* [98,99]. Another mechanism, utilized by microbiota, which can affect the invasion of a different enteric pathogen, *C. jejuni*, is the production of secondary bile acids which can block pathogenic mTOR signaling [100,101]. Microbiota can also regulate the activity and composition of ILCs via production of metabolic factors which can change cytokine production and activation of innate immune responses [102,103]. It is still largely unknown how signals from microbiota change transcriptional profiles of ILC subsets in the gut and how these changes affect tissue-specific functions and composition of ILCs upon infection. It was demonstrated that emergence of NCR^+^ ILC3s in the gut correlates with microbiota colonization, indicating that signals from microbiota may regulate ILC3s subpopulations and expression of NCRs by these cells [51]. In line with this, germ-free mice as well as antibiotic-treated mice displayed significant reduction in numbers of NCR^+^ RORγt^+^ ILCs [35,51,104]. Microbiota induces IL-7 and IL-2 production, important for ILC2s and ILC3s survival [13]. Consistently, germ-free or antibiotic-treated mice demonstrated reduced expression of epithelial-derived IL-7 in the intestine [94]. Another study showed that microbiota-induced expression of IL-7 stabilized RORγt expression in NCR^+^ RORγt^+^ ILCs as the transition of RORγt^+^ ILC3s to ILC1-like phenotype was inhibited in germ-free mice compared to conventionally colonized mice [35]. Thus, these studies support the idea that IL-7-dependent signals from microbiota prevent ILC3 to ILC1 plasticity via maintaining type 3 phenotype. Interestingly, during plastic transformation, NCR^+^ ILC3s downregulate IL-7R, making them less responsive to IL-7, and therefore leading to reduced RORγt expression [105]. Recently it was found that IL-7R expression on ILC3s is dependent on c-Maf as c-Maf deficiency led to reduced IL-7R expression and abrogated ILC3 to ILC1 transition [39]. Further research is necessary to determine how IL-7 production is regulated by microbiota and how it affects ILC phenotypes.

Transcriptional profiling of ILCs provided insight into the link between microbiota composition and ILC subsets [75]. Although an overall transcriptional identity of ILC subsets was not affected by antibiotics treatment, ILC1 and ILC2 expression profiles were changed by microbiota depletion [75]. Moreover, depletion of microbiota led to upregulation of ILC3-specific genes in both ILC1 and ILC2 subsets, indicating that ILC subsets could differentially respond to microbiota signals and that microbiota-derived signals maintain the ILC3 phenotype [75]. However, more studies are necessary to determine how pathogens change the host microbiome and whether interaction between pathogens and microbiome affect ILC plasticity.

## 5. Role of Intestinal Hypoxia in ILC Plasticity

Acute and chronic intestinal inflammation results in tissue damage that can be exacerbated by hypoxia [106]. During oxygen deprivation hypoxia-inducible factor-1α (HIF-1α) is activated to help cells adapt to low levels of oxygen [106,107]. HIF-1α activation correlates with protection against intestinal inflammation during bacterial infections and in a chemically-induced colitis model [108,109,110,111]. Thus, *Salmonella* infection leads to stabilization and activation of HIF-1α in epithelial cells which is associated with a protective immune response [110,112]. Proinflammatory cytokines IL-1β and IL-23 upregulate the expression of HIF-1α in ILC3s [113]. Moreover, the induction of HIF-1α is accompanied by upregulation of RORγt, suggesting that HIF-1α promotes ILC3 phenotype by inducing RORγt expression [113]. This was further supported by a recent study which showed that hypoxia activates HIF-1α, which in turn directly induces RORγt expression in ILC3s, similarly to previously reported regulation of RORγt by HIF-1α in T cells [114,115,116]. It was further demonstrated that activation of ILC3s during *Citrobacter rodentium* and *Clostridioides difficile* infections resulted in metabolic changes that are associated with activation of the mammalian target of rapamycin (mTOR) [113,114] (Figure 3). mTOR is a protein kinase operating through two protein complexes mTORC1 (RAPTOR—regulatory-associated protein of mTOR) and mTORC2 (Rictor—rapamycin-insensitive companion of mTOR) [117]. Activation of mTORC1 enhances expression of HIF-1α, which in turn induces expression of cell survival genes during hypoxia [117,118]. mTORC1 has been shown to promote ILC3 proliferation and IL-17/IL-22 production during *Citrobacter rodentium* infection [113]. ILC3 activation by IL-1β/IL-23 cytokines leads to generation of reactive oxygen species (ROS) in mitochondria [113]. In turn, mitochondrial ROS support prolonged activation of HIF-1α, further amplifying IL-22 and IL-17 production by ILC3s [113] (Figure 3). Interestingly, mTOR signaling controls ILC3 homeostasis in the small intestine but not in the colon, probably due to higher mTOR signaling in ILC3s in the small intestine compared to the colon [119]. Moreover, IFNγ production by ILC3s was dependent on mTORC1 and mTORC2 signaling in the small intestine, whereas IL-22 production by ILC3s was only dependent on mTORC1, suggesting distinct mechanisms of IL-22 and IFNγ activation in different ILC3 subsets [119]. 

Loss of HIF-1α in all RORγt-expressing ILCs resulted in decreased levels of IL-22 and IL-17 and reduced numbers of ILC3s; however, ablation of HIF-1α in NCR^+^ ILC3s led to increased IL-22 production and downregulation of IFNγ in the small intestine [97,114]. These results indicate that the contribution of HIF-1α to ILC plasticity is cell-type- and environment-dependent. Thus, HIF-1α expression in NCR^+^ ILC3s contributes to ILC3→ILC1 plasticity, whereas expression of HIF-1α in NCR^−^ ILC3 subsets supports ILC3 maintenance.

Since microbiota-derived metabolites can control HIF-1α-dependent expression of IL-22 in the gut [97,113,114,120], it is possible that microbiota composition regulates ILCs plasticity. Therefore, mechanisms of microbiota-dependent regulation of HIF-1α and its role in supporting ILC1↔ILC3 plasticity in the gut remain to be fully evaluated.

## 6. NK to ILC1 Plasticity in Response to *T. gondii*

*Toxoplasma gondii* (*T. gondii*) infection is associated with systemic dissemination of the parasite to different tissues such as spleen, lungs, brain and liver [32,121]. While both conventional NK cells and ILC1s are known as main cellular sources of IFNγ for control of *T. gondii* infection in the gut, recent study revealed an important role of ILC1-mediated immune response in early control of cerebral *T. gondii* infection [122]. NK cells and ILC1s are phenotypically close. Moreover, inflammatory conditions can promote changes in ILC1s and NK cells leading to induction of populations of ILC1s and NK cells with phenotypes which are very similar to each other but distinct from the phenotypes of cells under steady state conditions, making the classification of NK cells and ILC1s more challenging [26,32]. In the context of inflammation or tissue injury, ILC1s can acquire phenotypes largely overlapping with NK cells [32,33,34]. A recent study identified infection-mediated changes of ILC1s and NK cells in response to *T. gondii* [32] (Figure 4). Induction of IFNγ-mediated immune response was accompanied by Ly6C^hi^ monocytes appearance [122,123]. Furthermore, *Toxoplasma* infection resulted in a decrease of NK cell numbers in the spleen with the concomitant increase of cell populations resembling ILC1s in naïve mice [32]. Additionally, it was found that Eomes^+^ NK cells downregulated Eomes expression during *T. gondii* infection and acquired an ILC1-like phenotype (Figure 4). Interestingly, expansion of ILC1-like cells occurred independently from parasite replication as a *Toxoplasma* strain which cannot replicate in the host induced comparable accumulation of ILC1-like cells [32]. Moreover, *T. gondii*-mediated expansion of ILC1-like cells was sustained in the spleen and lungs even after the resolution of infection [32].

Although ILC1s are mainly considered to be tissue-resident cells, a *T. gondii* induced ILC1-like cell population was also found in the blood, indicating that these cells are circulating [32]. In contrast to ILC1s, conventional NK cells are migratory cells, which can enter lymph nodes from the blood in a CD62L^−^ and CCR7-dependent manner [124,125,126]. Subsequently, the ILC tissue residency concept has been challenged by the study describing ILC1s that can migrate from lymphoid and nonlymphoid tissues in a similar manner to cNK cells [127]. Interestingly, *T. gondii*-induced ILC1-like cells express Neuropilin-1 (NRP-1) that acts as a coreceptor for class 3 semaphorins (Sema3), VEGF and TGFβ [32,128]. It is established that NRP-1 expression on ILC3s facilitates their migration to lymphoid tissues through high endothelial venules [79,129]. Moreover, VEGF-VEGFR2 signaling serves as a chemotactic factor for NRP-1^+^ ILC3s to migrate to the sites of inflammation [79,129]. In line with these reports, another study showed that NRP-1 expression on ILC2s is regulated by the tissue environment [128]. ILC2s upregulate NRP-1 when they enter the lungs and downregulate its expression when they leave the lungs, suggesting that tissue-specific signals control expression of NRP-1 and recruitment of ILCs to the sites of inflammation [128]. Consistent with the expression of genes associated with cell migration and the presence of ILC1-like cells in the circulation, it is possible that *T. gondii* infection induces changes in the phenotype of NK cells in the spleen and liver, allowing these cells to disseminate to other organs with a high rate of inflammation, where they can induce protective IFNγ-mediated immune response. ILC1-like cells induced by *Toxoplasma* infection express classical NK markers such as Ly6C, KLRG1and CD11b [32]. Upon activation, these ILCs produce IFNγ, thereby contributing to pathogen control [32]. Although IFNγ produced by ILC1s is important for protection against the parasite at the onset of the infection, NK cells and T cells are critical for protection at the later stages of infection [130,131,132,133]. Interestingly, another study showed that chronic *Toxoplasma* infection facilitated the development of a unique activated NK cell population which can negatively regulate CD8^+^ T cell immune responses, resulting in persistent parasite infection of the brain [134]. Given the heterogeneity of NK phenotypes in different organs upon *T. gondii* infection and NK to ILC1 plasticity, it is possible that similar conversion occurs in mucosal tissues, such as gut. Since *T. gondii* disseminates to other organs and ILC1s are critical for IFNγ-dependent control of infection in the intestine, it would be interesting to test whether these cells undergo plasticity in the gut, or they migrate from other inflammatory sites to facilitate host protection.

## 7. ILC Plasticity in Response to Genital Tract Infections 

Recent studies showed the presence of all ILC subsets in the reproductive tract and fetal tissues [135,136,137]. Moreover, the role of ILCs in genital tract infections was recently demonstrated in a *Chlamydiae* infection model [53,54,55,138,139]. IFNγ-producing ILCs are protective against *Chlamydiae* [53,54,55,138,139]. Since activated T-bet^+^ RORγt^+^ ILC3s produce IFNγ and adoptive transfer of RORγt^+^ ILC3s protected mice from intestinal colonization with *Chlamydia* mutant, it was proposed that ILC plasticity can occur in response to *Chlamydia* infection [51,54]. In line with these reports, another study showed that all three subsets of ILCs were present in mouse female genital tract during chlamydial infection; however, infection led to expansion of ILC1s and NK cells with slight changes in ILC2s but no expansion of ILC3s [53]. Further cell fate mapping experiments showed that ILC1 subset included cells with a previous history of RORγt expression [53], supporting the idea that *Chlamydia* infection facilitates ILC3 to ILC1 plasticity (Figure 2). Further studies will help to identify the functional relevance of ILC plasticity in the genital tract.

*Chlamydiae* infection is associated with changes in the vaginal microbiome that can impact IFNγ-mediated immune response [140,141]. Accumulating evidence suggests that dissemination of *Chlamydia* in the lower genital tract can affect species dominance and abundance in upper female reproductive tract [142,143]. Since ILCs are more abundant in the oviducts of the mouse female genital tract [53], it would be interesting to investigate how the different environment throughout the genital tract affects ILC function and phenotypes. ILC plasticity in other genital tract infections besides *Chlamydiae* such as *Neisseria gonorrhoeae* and *Treponema pallidum* have not been demonstrated yet. The role of ILCs in genital tract infections of human patients also awaits future studies. As study of human ILCs plasticity is challenging, humanized mouse models can provide further insights into this mechanism.

## 8. ILC Plasticity in Response to Lung Infections

Lungs are constantly exposed to bacteria, viruses as well as noxious agents (small dust, pollen particles etc.) from the environment that can cause pulmonary disease. Although in homeostatic conditions lungs harbor B, T cells and myeloid cells to support the barrier function, growing evidence suggests that ILCs also contribute to the lung homeostasis and tissue repair after infection [20] (Figure 5). All major ILC subsets were detected in the airways and lungs with ILC2s being a predominant cell population in the mouse lungs in homeostatic conditions, whereas ILC3s are prevalent in human lungs in pulmonary diseases [20,144]. Furthermore, studies in mice and humans have reported that ILC2s can also adopt alternative fates and convert to IL-17^+^ ILC3-like or IFNγ^+^ ILC1-like cells under inflammatory conditions [31,42,43,45,46,48,89]. Although ILCs are mainly considered as tissue-resident cells, emerging data suggest that inflammation can induce migration of some ILC populations, including ILC precursors (ILCP) to the sites of infection or inflammation where they can adopt alternative fates or differentiate into tissue-resident ILC subsets based on the tissue-specific signals [126,127,145,146]. A recent study showed that ILCPs in the blood can give rise to all ILC subsets [147,148]. ILCPs can migrate from the blood to the tissues to replenish tissue resident ILCs during inflammation [147,149]. In addition to ILCP, the lungs have a unique tissue-resident population of IL-18R^+^ ILCs with an intermediate phenotype between progenitor and mature ILCs [44,70]. These cells display high expression of TCF-1 (T cell factor-1) which controls the development of ILC progenitor cells at early stages [88,148,150]. Furthermore, IL-18R^+^ ILCs express Arginase 1 (Arg1) which is considered as a selective marker of tissue-resident ILC2s [44,146]. While mature ILCs usually do not proliferate, pulmonary IL-18R^+^ ILCs can proliferate and give rise to ST2^+^IL-18R^−^ ILCs, ST2^−^IL-18R^+^ ILCs and ST^+^IL-18R^+^ ILCs at a steady state [70].

IL-18R expression has been detected in bone-marrow progenitor cells [151,152]. Furthermore, IL-18 produced during inflammation can reduce hematopoietic stem cell differentiation [152,153]. Additionally, a recent work suggests that IL-18R signaling is dispensable for ILC development but inhibits differentiation of ILCPs [151]. Since IL-18R^+^ ILCs are highly proliferative, their responsiveness to IL-18 may be an important mechanism to prevent excessive cell proliferation during inflammation [44,70]. In fact, helminth *Nippostrongylus brasiliensis* (*Nb*) can drive maturation of IL-18R^+^ ILCs to ILC2s in the lungs at early stages of infection [70]. However, further studies are needed to determine the role of IL-18R signaling in regulating tissue-resident progenitor ILCs in the lungs.

Recent studies demonstrated that *Mycobacterium tuberculosis* (*Mtb*) infection induces ILCs accumulation in the lungs [44,154]. Furthermore, *Mtb* can induce IL-18R^+^ ILCs in the lungs [44]. It was also found that *Mtb* infection changes ILC composition, leading to accumulation of protective IFNγ-producing ILC1s and IL-17 producing ILC3s [154] in parallel with reduction of ILC2s [44]. Lung IL-18R^+^ ILCs are mainly ILC2-like cells, characterized by GATA3 and ST2 (IL-33 receptor) expression [44]. Moreover, over the course of infection, IL-18R^+^ ILCs start to express T-bet with the concomitant IFNγ production to acquire a ILC1-like phenotype [44] (Figure 5). Prior studies showed that IL-1β + IL-12 treatment of purified ILC2s in vitro induced IL-18R along with T-bet and IL-12R expression with concomitant reduction of ST2 and GATA3 [43,48]. The expression of IL-12Rβ1 is required for human ILC2s to acquire ILC1-phenotype [31]. Therefore, it is possible that IL-1β produced during *Mtb* infection potentiates the appearance of IL-18R^+^ ILCs by inducing expression of IL-12R and making these cells more responsive to IL-12 thereby driving ILC2→ILC1 plasticity (Figure 5).

Although IFNγ is important for protection against *Mtb*, a recent study suggests that IL-22 and IL-17 producing ILC3s can also contribute to protective immune response, potentially via maintenance of inducible bronchus-associated lymphoid tissue (iBALT) [154]. Therefore, it is tempting to speculate that *Mtb* can also promote ILC3→ILC2 plasticity (Figure 5). However, this possibility has not been experimentally tested yet.

Recent studies indicate that influenza virus infection can also induce ILC2→ILC1 plasticity [48]. Influenza infection led to the loss of GATA3 and ST2 expression in ILC2s and subsequent increase of IL-18R^+^T-bet^+^ ILCs [48]. Furthermore, the majority of IFNγ-producing IL-18R^+^ ILCs were able to proliferate after infection and displayed an immature phenotype [48,70]. In contrast to IL-18R^+^ ILCs, mature IL-18R^−^ ILC2s were unable to give rise to ILC1-like cells [44]. Thus, these studies demonstrate that the ability of IL-18R^+^ tissue-resident ILC precursors to undergo plasticity allows them to quickly induce a protective immune response against respiratory pathogens. 

Recent studies identified transcription factor Batf as an additional regulator of ILC2 identity [146,155,156]. Although Batf is known to promote stabilization of ILC3 phenotype [37], it can also participate in the maintenance of ILC2 identity [146,155,156]. The lung population of ILC2s express Batf at a steady state [146]. Influenza infection can further increase Batf expression in ILC2s by epithelial-derived IL-33 [156]. In turn, Batf promotes the stability of the ILC2 phenotype by inducing expression of IL-33R (ST2), making cells more responsive to IL-33 [156]. Batf deficiency in all ILCs led to increased number of pathogenic IL-17- and IFNγ-producing ILC2s as well as neutrophils, macrophages and monocytes in the lungs [156]. Batf deficiency also resulted in reduced GATA3 expression in ILC2s, potentially via inhibition of ILC2-characteristic genes, such as *Il5*, *Il13*, *Areg* [156]. 

In line with these studies, ILC2→ILC3 plasticity was shown in response to *Candida albicans* infection [50]. Furthermore, human dermal-derived ILC2s can acquire ILC3-like phenotype and produce IL-17 during infection [46]. Interestingly, analysis of dermal RORγt^+^ ILC2s revealed lower levels of Batf expression [46] implicating the role of Batf in the maintenance of human ILC2 phenotype. Thus, these studies suggest that infection induces context-dependent regulation of Batf expression to promote ILC plasticity. 

Notch signaling in the lungs can also contribute to ILC2→ILC3 plasticity [89]. Inflammatory conditions induce emergence of IL-13/IL-17 producing ILC2s. Both Notch1 and Notch2 receptors are required for these ILC2s to emerge [89]. Activation of Notch receptors initiates cleavage of Notch intracellular domain from the membrane and its translocation to the nucleus. Notch binds to transcription factor CSL (also known as RBP-Jκ) and coactivator protein Mastermind homologue (MAML1-3) to form transcription activation complex [157]. This transcriptional complex directly binds to *Rorc* promoter and induces expression of RORγt leading to the production of IL-17 by ILC2s in the lungs [89]. This is in line with the study demonstrated that Notch signaling stabilized RORγt expression, thus promoting ILC3 differentiation [148]. Since Notch signaling is critical for ILC2→ILC3 plasticity, the distinct availability of Notch ligands in different tissues could be an important factor determining of ILCs composition.

Recent studies indicate that ILC subsets are dysregulated in patients with severe COVID-19 [60,158,159]. Furthermore, it was proposed that lower abundance of homeostatic ILCs correlates with the severity of disease [59]. However, the role of ILC plasticity in the pathogenesis of SARS-CoV-2 induced lung and intestinal disease remains unknown.

## 9. Conclusions and Future Directions

Recent advances in ILC biology extended our understanding of the functions of ILCs in host defense, tissue damage and chronic inflammation. It is established now that ILCs not only participate in protective responses but can also contribute to inflammation and tissue damage. A growing body of evidence support the concept that ILC phenotypes are flexible and that pathogen invasion or inflammatory conditions change ILC phenotypes and functions according to local tissue microenvironment cues. Recent studies applying transcriptomic and single cell analyses demonstrated the phenotypical heterogeneity of ILCs under physiological and pathological conditions. However, how ILCs are regulated under infection-driven inflammatory conditions remain poorly understood. The precise contribution of tissue environment and unique pathogen-driven signals in ILC plasticity remain to be further studied. Despite the ability of ILCs to revert their phenotype after conversion, it is still unclear whether reverse transdifferentiation occurs after resolution of pathogen-induced inflammation. ILC plasticity has been demonstrated in response to several pathogens (Table 1). However, the ILC plasticity and its significance in response to many other pathogens has not been described yet. It is also unclear whether pathogens can exploit ILC plasticity to their own benefit to help avoid host protective responses. 

The majority of studies of ILC plasticity in vivo are based on mouse models. However, our knowledge of ILC plasticity mechanisms in humans is mainly based on in vitro studies and correlation studies of ILC subsets in healthy and inflamed human tissues. It is yet unclear whether human ILCs undergo the same plasticity after pathogen invasion as described in mice. Multiple studies showed the importance of microbiome and diet in regulation of ILC effector functions, however, how microbiome and metabolites impact ILC diversity and plasticity in different tissues remains to be determined.

Recent studies found that changes in human ILC populations are associated with age and with increased body mass index [78]. As children, older people and immunodeficient patients are among the most susceptible to infectious diseases due to immature or weakened immune system, it is critical to determine how the plasticity of ILCs controls the activation of host immune response upon pathogen invasion in these individuals. Another important factor worth further discussion in the context of ILC plasticity, is sexual dimorphism. Recent studies suggested that sex hormones can regulate ILC2 function in the lungs [160,161]. Men have reduced numbers of progenitor and mature ILC2s in the peripheral tissues compared with women [160]. Androgen receptor signaling negatively regulates lung inflammation which correlates with increased prevalence of asthma in women [160,161]; however, the role of sex hormones in driving ILC plasticity in response to infections is yet to be elucidated. Emerging evidence suggests that ILCs can regulate the function of adaptive immune cells. It remains to be determined how adaptive immune cells regulate ILC plasticity. Finally, identification of new pathways and molecular mechanisms that control ILC functions and plasticity could uncover more specific and effective therapeutic targets for human diseases where ILCs become dysregulated.

## Figures and Tables

**Figure 1 microorganisms-11-00461-f001:**
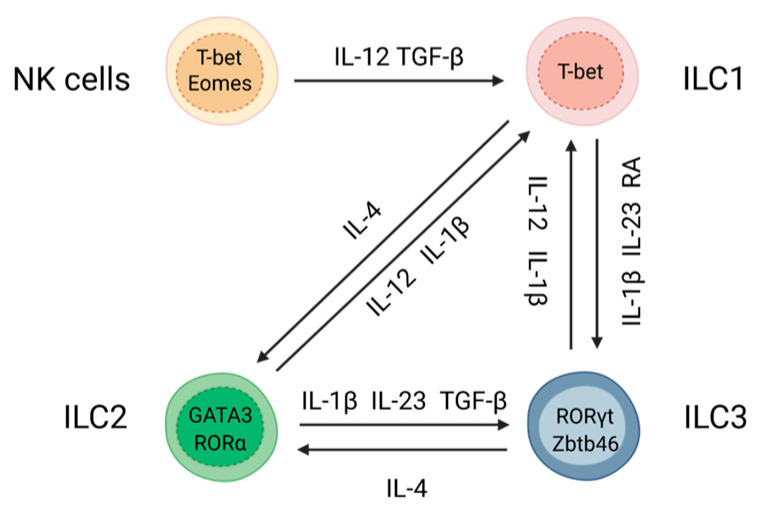
Plasticity among ILC subsets. ILCs plasticity is driven by specific cytokines and the balance between key transcription factors.

**Figure 2 microorganisms-11-00461-f002:**
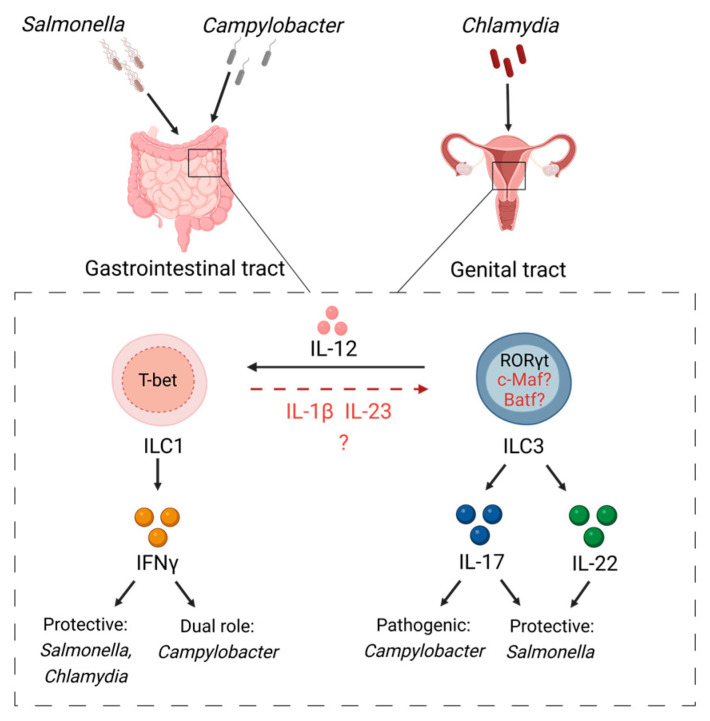
ILC plasticity in response to gut and genital tract infections. *Salmonella* and *Campylobacter* species induce IL-12 production by accessory cells to drive conversion of ILC3s toward IFNγ-producing ILC1s in the intestine by downregulating RORγt and upregulating T-bet expression. Similarly, *Chlamydia* drives ILC3→ILC1 conversion to initiate IFNγ-mediated protective responses.

**Figure 3 microorganisms-11-00461-f003:**
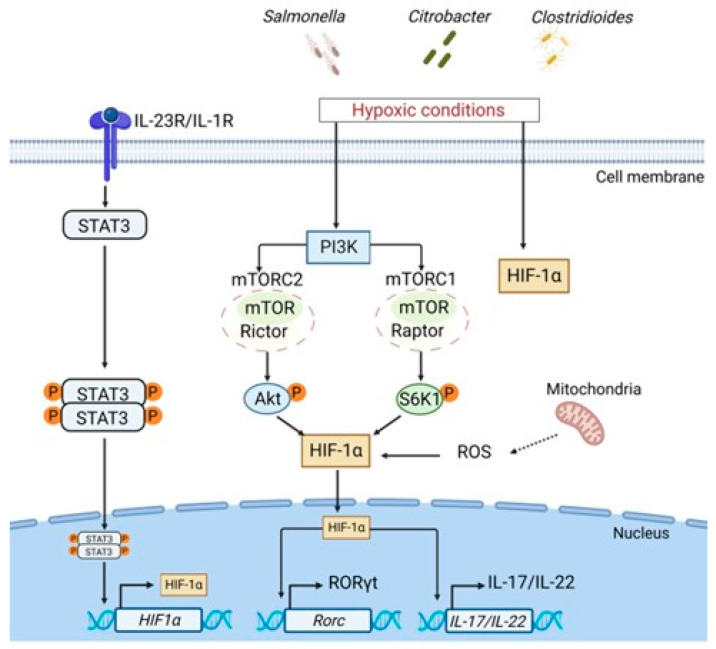
Role of hypoxia in regulation of ILC3 responses. Bacterial infections, such as *Salmonella*, *Citrobacter* or *Clostridioides* promote intestinal hypoxia. Under homeostatic conditions HIF-1α is expressed at low levels. During hypoxia, HIF-1α is activated and stabilized via STAT3, PI3K-mTOR signaling pathways and reactive oxygen species (ROS) produced in mitochondria. HIF-1α translocates to the nucleus and induces transcription of RORγt and IL-17/IL-22 cytokines to protect against bacterial pathogens.

**Figure 4 microorganisms-11-00461-f004:**
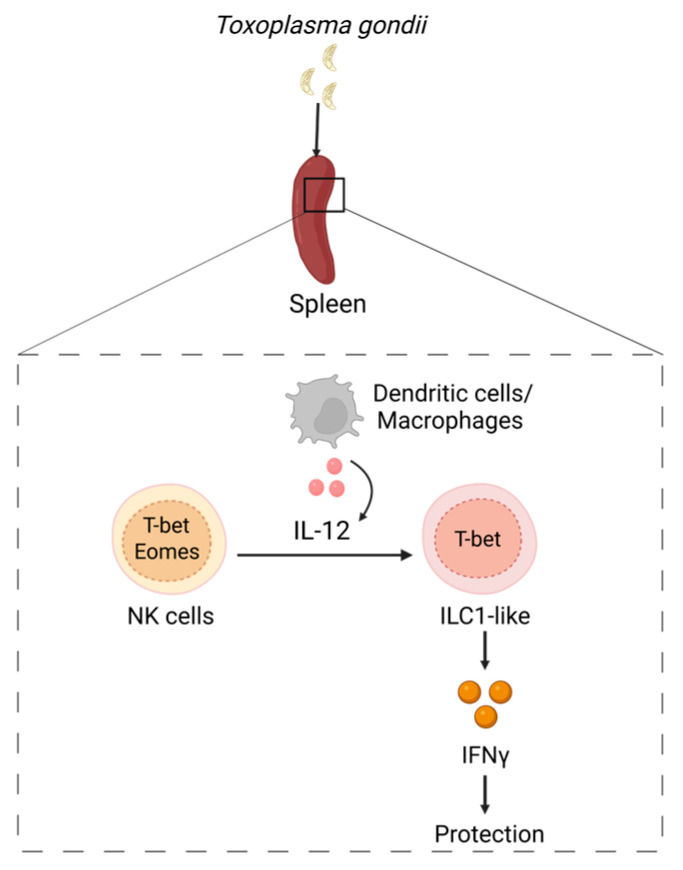
ILC plasticity in response to *T. gondii*. *T. gondii* activates dendritic cells and macrophages to produce IL-12 which drives NK→ILC1 plasticity in the spleen, liver and lungs by downregulating expression of Eomes. IFNγ is required for protection against *T. gondii*.

**Figure 5 microorganisms-11-00461-f005:**
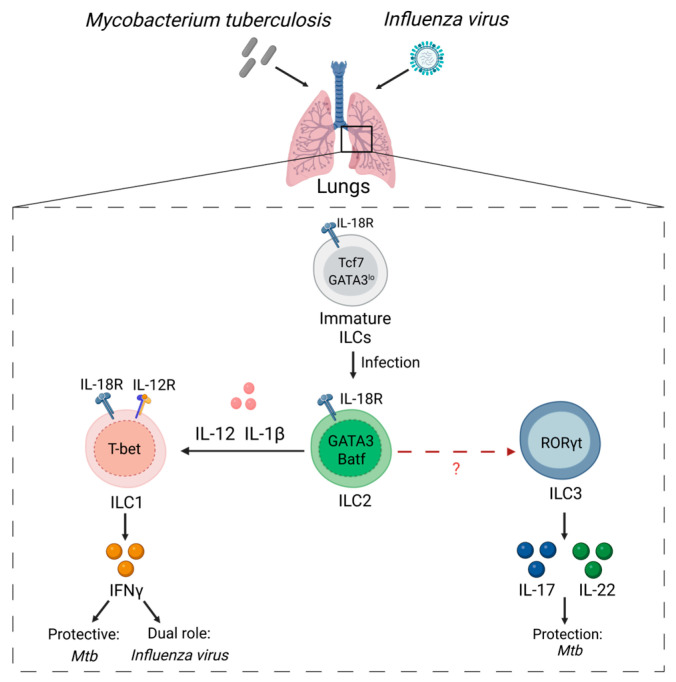
ILC plasticity in response to lung infections. In steady state lungs contain tissue-resident immature IL-18R^+^TCF1^+^ GATA3^lo^ ILCs which can differentiate into ILC2s. Respiratory infections facilitate maturation of IL-18R^+^ ILCs. Type 1 inflammatory cytokines (IL-12+IL-1β) drive ILC2→ILC1 plasticity by downregulating GATA3 and upregulating T-bet. T-bet induces production of IFNγ by ILC1s. Additionally, IL-22 and IL-17 produced by ILC3s support protective responses via iBALT maintenance.

**Table 1 microorganisms-11-00461-t001:** ILCs plasticity in response to pathogens.

ILC Plasticity	Tissue	Pathogen	Type of Pathogen	Main Cytokine and Function
NK→ILC1	Spleen, lungs, liver	*Toxoplasma gondii*	Protozoan parasite	Protective IFNγ [32]
ILC2→ILC1	Lungs	*Mycobacterium tuberculosis*	Intracellular bacteria	Protective IFNγ [44]
	Lungs	*Influenza* virus	Respiratory virus	Protective/pathogenic IFNγ [48,49]
ILC2→ILC3	Skin, Tongue	*Candida albicans*	Fungus	Protective IL-17 [46,50]
	Intestine, lungs	*Nippostrongylus brasiliensis*	Helminth	Protective/pathogenic IL-17 [50]
ILC3→ILC1	Intestine	*Salmonella typhimurium*	Intracellular bacteria	Protective/pathogenic IFNγ [51]
	Intestine	*Campylobacter jejuni*	Intracellular bacteria	Pathogenic/protective IFNγ [52]
	Genital tract	*Chlamydia* spp.	Intracellular bacteria	Protective IFNγ [53,54,55]
ILC plasticity not yet determined	Lungs	SARS-CoV-2	Respiratory virus	Pathogenic/protective IFN type I-III, TNF, IL-1b, CCL2 [56,57,58,59,60]
	Intestine	*Yersinia enterocolitica*	Intracellular bacteria	Protective IFNγ [61]
	Spleen, liver	*Plasmodium* spp.	Protozoan parasite	Protective IL-4/IL-5/IL-13 [62]; Protective IFNγ [63,64]
	Intestine	*Clostridium difficile*	Extracellular bacteria	Protective IFNγ/IL-22 [65]
	Intestine	HIV-1	Virus	Protective IL-22 [66,67,68,69]

## Data Availability

No new data were created in this study. All the data reported in this review were found in original articles cited in the text and Figures. Literature used to inform the text of this article was selected from https://pubmed.ncbi.nlm.nih.gov (accessed on 5 February 2023) from the National Library of Medicine. Full length manuscripts and Communications published in the English language between 2000 and 2022 were selected.

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
