# Peer review of "Innate Lymphoid Cell Plasticity in Mucosal Infections"

_microorganisms, 2023, doi:10.3390/microorganisms11020461_

Round 1

Reviewer 1 Report

The review examines previous studies and compiles data and evidence regarding the plasticity of Innate Lymphoid cells in mucosal infections. The structure of the text is excellent and well-written. The images help to summarize the information and what the authors are informing. 

It is a good review and I would suggest the author the discussion of the role of sexual dimorphism in the ILCs' plasticity in future studies. Indirectly, some aspects were addressed when the genital tract was discussed. 

Author Response

Reviewer 1

 Comments and Suggestions for Authors

The review examines previous studies and compiles data and evidence regarding the plasticity of Innate Lymphoid cells in mucosal infections. The structure of the text is excellent and well-written. The images help to summarize the information and what the authors are informing. 

We thank the Reviewer for the positive summary of our work and helpful comments. We made suggested changes which significantly improve our manuscript.

  1. It is a good review and I would suggest the author the discussion of the role of sexual dimorphism in the ILCs' plasticity in future studies. Indirectly, some aspects were addressed when the genital tract was discussed. 

We thank the Reviewer for this suggestion. The section “Conclusions and future directions” now includes the discussion of the role of sexual dimorphism in the ILCs' plasticity in the revised manuscript (lines 620-627).

Reviewer 2 Report

The authors provide a comprehensive view of the plasticity of the ILCs in different pathological conditions. Overall the review is well written and summarizes the current knowledge in this field.  The figures also improve the clarity of the review. It is also interesting that the authors discuss the plasticity of the ILCs in different tissues.

I would suggest adding the following studies that could be relevant for the review.

Lim, A. I. et al. IL-12 drives functional plasticity of human group 2 innate lymphoid cells. J. Exp. Med. 213, 569–583 (2016)

Zhang, K. et al. Cutting edge: Notch signaling promotes the plasticity of group-2 innate lymphoid cells. J. Immunol. 198, 1798–1803 (2017)

I would like to point out that REF 43 and 47 are the same study. 

43. Corral, D.; Charton, A.; Krauss, M. Z.; Blanquart, E.; Levillain, F.; Lefrançais, E.; Sneperger, T.; Girard, J.-P.; Eberl, G.; Poquet, 735 Y.; Guéry, J.-C.; Arguello, R. J.; Hepworth, M. R.; Neyrolles, O.; Hudrisier, D., Metabolic control of type 2 innate lymphoid 736 cells plasticity toward protective type 1-like cells during <em>Mycobacterium tuberculosis</em> infection. bioRxiv 2021, 737 2021.01.19.427257.

47. Corral, D.; Charton, A.; Krauss, M. Z.; Blanquart, E.; Levillain, F.; Lefrançais, E.; Sneperger, T.; Vahlas, Z.; Girard, J. P.; Eberl, 747 G.; Poquet, Y.; Guéry, J. C.; Argüello, R. J.; Belkaid, Y.; Mayer-Barber, K. D.; Hepworth, M. R.; Neyrolles, O.; Hudrisier, D., 748 ILC precursors differentiate into metabolically distinct ILC1-like cells during Mycobacterium tuberculosis infection. Cell 749 reports 2022, 39, (3), 110715.

Author Response

Reviewer 2

 Comments and Suggestions for Authors

The authors provide a comprehensive view of the plasticity of the ILCs in different pathological conditions. Overall, the review is well written and summarizes the current knowledge in this field.  The figures also improve the clarity of the review. It is also interesting that the authors discuss the plasticity of the ILCs in different tissues.

We thank the Reviewer for the positive comments.

  1. I would suggest adding the following studies that could be relevant for the review.

Lim, A. I. et al. IL-12 drives functional plasticity of human group 2 innate lymphoid cells. J. Exp. Med. 213, 569–583 (2016)

Zhang, K. et al. Cutting edge: Notch signaling promotes the plasticity of group-2 innate lymphoid cells. J. Immunol. 198, 1798–1803 (2017)

We thank the Reviewer for this excellent suggestion. We added these two studies to the revised text (lines 218-220, 537-539, 573-585)

  1. I would like to point out that REF 43 and 47 are the same study. 
  2. Corral, D.; Charton, A.; Krauss, M. Z.; Blanquart, E.; Levillain, F.; Lefrançais, E.; Sneperger, T.; Girard, J.-P.; Eberl, G.; Poquet, 735 Y.; Guéry, J.-C.; Arguello, R. J.; Hepworth, M. R.; Neyrolles, O.; Hudrisier, D., Metabolic control of type 2 innate lymphoid 736 cells plasticity toward protective type 1-like cells during <em>Mycobacterium tuberculosis</em> infection. bioRxiv 2021, 737 2021.01.19.427257.

  1. Corral, D.; Charton, A.; Krauss, M. Z.; Blanquart, E.; Levillain, F.; Lefrançais, E.; Sneperger, T.; Vahlas, Z.; Girard, J. P.; Eberl, 747 G.; Poquet, Y.; Guéry, J. C.; Argüello, R. J.; Belkaid, Y.; Mayer-Barber, K. D.; Hepworth, M. R.; Neyrolles, O.; Hudrisier, D., 748 ILC precursors differentiate into metabolically distinct ILC1-like cells during Mycobacterium tuberculosis infection. Cell 749 reports 2022, 39, (3), 110715.

We thank the Reviewer for pointing out to this oversight and we corrected it in the revised manuscript.

Reviewer 3 Report

The manuscript submitted by Korchagina et al. provides an extensive review of the role of innate lymphoid cells in mucosal infections and the phenomenon of cellular plasticity that occurs in this context.

The authors have presented a well-written, well-structured article with a large number of references.

I believe that the authors should only attend to minor revisions for the publication of this article in the journal Microorganisms.

-The abstract of the manuscript is too general, the authors should include a more concrete conclusion within this section.

-A paragraph indicating the methodology of the searches conducted is missing: search terms, limit of years.... I believe that this small section could greatly enrich the manuscript.

-The figure legends are excessively informative. The legends should be more concise. Please refer to the information in the main text. Maybe different panels (Panel A, Panel B…) within the figures can help to refer the figures in the text. 

Author Response

Reviewer 3

Comments and Suggestions for Authors

The manuscript submitted by Korchagina et al. provides an extensive review of the role of innate lymphoid cells in mucosal infections and the phenomenon of cellular plasticity that occurs in this context.

The authors have presented a well-written, well-structured article with a large number of references.

We are thankful to the Reviewer  for the thoughtful evaluation of our work. Our detailed point-by-point responses to Reviewer’s suggestions are below.

I believe that the authors should only attend to minor revisions for the publication of this article in the journal Microorganisms.

  1. The abstract of the manuscript is too general, the authors should include a more concrete conclusion within this section.

We added more detailed conclusions to the revised manuscript (line 16-20).

  1. A paragraph indicating the methodology of the searches conducted is missing: search terms, limit of years.... I believe that this small section could greatly enrich the manuscript.

We added section of “Literature search” in the revised manuscript (lines 639-641)

  1. The figure legends are excessively informative. The legends should be more concise. Please refer to the information in the main text. Maybe different panels (Panel A, Panel B…) within the figures can help to refer the figures in the text. 

We updated Figure legends in the revised manuscript to make them more concise (Figure 2: lines 185-193; Figure 3: lines 349-360; Figure 5: lines 505-516).
